scAnt—an open-source platform for the creation of 3D models of arthropods (and other small objects)

http://orcid.org/0000-0003-1012-6646 Plum Fabian
Labonte David d.labonte@imperial.ac.uk
Department of Bioengineering, Imperial College London , London , UK
Gillespie Joseph
Electronic publication date: 2021 Apr 12
Publication date: 2021
Volume: 9
Electronic Location ID: e11155
Received 2020 Dec 27; Accepted 2021 Mar 4
Copyright: © 2021 Plum and Labonte
Copyright year: 2021
Copyright holder: Plum and Labonte
License: This is an open access article distributed under the terms of the Creative Commons Attribution License, which permits unrestricted use, distribution, reproduction and adaptation in any medium and for any purpose provided that it is properly attributed. For attribution, the original author(s), title, publication source (PeerJ) and either DOI or URL of the article must be cited.
License URL: https://creativecommons.org/licenses/by/4.0/

Keywords: Photogrammetry, Morphometry, Zoology, Macro imaging, 3D, Digitisation

Funding: Imperial College’s President’s PhD Scholarship European Research Council (ERC) European Union’s Horizon 2020 851705 This study was funded by the Imperial College’s President’s PhD Scholarship (to Fabian Plum) and is part of a project that has received funding from the European Research Council (ERC) under the European Union’s Horizon 2020 research and innovation programme (Grant agreement No. 851705, to David Labonte). The funders had no role in study design, data collection and analysis, decision to publish, or preparation of the manuscript.

==============================
We present scAnt, an open-source platform for the creation of digital 3D models of arthropods and small objects. scAnt consists of a scanner and a Graphical User Interface, and enables the automated generation of Extended Depth Of Field images from multiple perspectives. These images are then masked with a novel automatic routine which combines random forest-based edge-detection, adaptive thresholding and connected component labelling. The masked images can then be processed further with a photogrammetry software package of choice, including open-source options such as Meshroom, to create high-quality, textured 3D models. We demonstrate how these 3D models can be rigged to enable realistic digital specimen posing, and introduce a novel simple yet effective method to include semi-realistic representations of approximately planar and transparent structures such as wings. As a result of the exclusive reliance on generic hardware components, rapid prototyping and open-source software, scAnt costs only a fraction of available comparable systems. The resulting accessibility of scAnt will (i) drive the development of novel and powerful methods for machine learning-driven behavioural studies, leveraging synthetic data; (ii) increase accuracy in comparative morphometric studies as well as extend the available parameter space with area and volume measurements; (iii) inspire novel forms of outreach; and (iv) aid in the digitisation efforts currently underway in several major natural history collections.

Introduction

The diversity of arthropods is unparalleled (Misof et al., 2014). Key institutions such as the Natural History Museum in London, the Smithsonian National Museum of Natural History, or the Australian National Insect Collection house upwards of ten million insect specimens, and grow continuously, so archiving part of this diversity (Mantle, La Salle & Fisher, 2012). Clearly, the utility of these collections hinges on the accessibility of the specimens. However, specimen access typically requires to be either physically present on-site, or for specimens to be posted, so reducing the practical value of the collections. This issue is particularly severe for rare and valuable specimens such as holotypes, which can be difficult to access despite their scientific importance. In recognition of these limitations, significant efforts have been underway to digitise natural history collections (Beaman & Cellinese, 2012; Blagoderov et al., 2012; Mantle, La Salle & Fisher, 2012; Nguyen et al., 2017, 2014; Hudson et al., 2015; Martins et al., 2015; Galantucci, Pesce & Lavecchia, 2016; Erolin, Jarron & Csetenyi, 2017; Ströbel et al., 2018; Galantucci, Guerra & Lavecchia, 2018; Qian et al., 2019; Brecko & Mathys, 2020). Such a cybertaxonomy has been predicted to revolutionise collaborative taxonomy, and fundamentally change formal and public taxonomic education (Zhang, Gao & Caelli, 2010; Wheeler et al., 2012). However, the vast majority of these efforts have focused on high-throughput capture of 2D images, and the convenient automated inclusion of metadata such as barcodes, labels etc. Photographs are doubtlessly useful, but by definition contain substantially less information than the original specimen, as they are restricted to a single image plane (Nguyen et al., 2014; Ströbel et al., 2018). Even obtaining simple 1D measurements from 2D images is error-prone, due to parallax errors and intra-observer variability, which is larger for measurements obtained from 2D photographs compared to 3D models (Ströbel et al., 2018; Qian et al., 2019; Brecko & Mathys, 2020). As a consequence, the “gold standard” for digitisation are photorealistic and anatomically accurate 3D models (Wheeler et al., 2012).

Perhaps the most promising method for the creation of 3D models is photogrammetric reconstruction, which retains colour information, and represents an excellent compromise between portability, price and quality (Mathys, Brecko & Semal, 2013; Brecko & Mathys, 2020). To the best of our knowledge, there exist two photogrammetry devices specifically designed for the creation of 3D models of arthropods (Nguyen et al., 2014; Ströbel et al., 2018). However, both systems rely on specialised hardware and commercial software for scanner control, image processing and 3D reconstruction, hampering widespread use.

Here, we address this limitation by introducing an open-source platform for the automated generation of 3D models of arthropods and small objects from a series of 2D images. This platform, scAnt, consists of (i) a low-cost scanner, built from generic structural and electronic components, (ii) an intuitive yet powerful Graphical User Interface (GUI) providing full control over (iii) a processing pipeline which combines several community-driven open-source applications to automate image capture and simultaneous image processing. scAnt runs on Windows and Linux operating systems, and includes full support for scanner control, image capture, image post-processing, as well as additional scripts to facilitate reconstruction. It is accompanied by a guide to manual mesh post-processing. All component drawings and assemblies are available on Thingiverse (thingiverse.com/evobiomech/designs, licensed under CC BY 4.0), and the manual and all code have been deposited on GitHub (github.com/evo-biomech/scAnt, made available under MIT License). Due to the combination of low-costs, open documentation and system flexibility, scAnt enables scientists and dedicated amateurs alike to create high-quality 3D models of arthropods, so contributing to a growing digital library which can be used for research, outreach and conservation.

Materials and Methods

scAnt is designed to automate the capturing and processing of 2D images from variable viewing angles. It is built entirely with generic hardware components, and exclusively leverages recent developments in open-source image processing software, so remaining affordable, accessible, and flexible.

With scAnt, 3D models are produced through a series of five steps, described in detail below: (i) mounting a pinned specimen in an illumination dome; (ii) configuring and conducting a “scan”, using a custom-made GUI; (iii) processing captured images into Extended Depth Of Field (EDOF) images and creating masks for each EDOF image; (iv) generating textured 3D meshes from the EDOF images; and (v) post-processing of the created mesh.

Scanner design

In the design of scAnt, we built on and benefited from several insights and innovations from previous studies: (i) We deploy focus stacking to overcome the limitations of single-focal plane images (Gallo, Muzzupappa & Bruno, 2014); (ii) We use a two-axis gimbaled system to maximise the number of possible viewing angles (Nguyen et al., 2014); (iii) We designed an illumination dome to achieve “flat” lighting, thus minimising specular reflections and other artefacts arising from variations in appearance with viewing angle (Ströbel et al., 2018).

All structural components of the scanner are fabricated via 3D printing and laser cutting, methods readily available in most laboratories and museums, but also accessible to the keen amateur. Technical drawings of all components, the assembly, and a complete parts list are available for download (thingiverse.com/evobiomech/designs, licensed under CC BY 4.0). The hardware costs for scAnt are approximately €200, and—due to the exclusive use of open-source software—the only remaining costs are related to the camera and lens (for our system, this added another €700). The total costs are hence between a factor of five to ten lower than for existing systems (~€5,000 for Nguyen et al., 2014; ~€8,000 for Ströbel et al., 2018).

In order to enhance hardware durability and to minimise print post-processing, we used PLA filament for all prints. An acrylic sheet, laser-cut from 4 mm thick acrylic sheets, serves as the base plate for the mounting of all printed elements, and for the routing of all wires (Fig. 1A).

Figure 1 Computer Aided Design (CAD) drawings and photograph of the assembled 3D scanner.

(A) Top-down drawing, including wiring. The stepper motors for each axis as well as their corresponding drivers are labelled X (horizontal axis, red wires), Y (vertical axis, orange wires) and Z (camera slider, yellow wires), respectively. Two end-stops (limit switches) are attached to the actuated camera rail to provide reference positions for the X and Z axes. Navy blue and maroon wires indicate connections from the 12 V power supply to the stepper drivers and LEDs, respectively. Blue wires indicate USB cables, connecting stepper drivers and camera to the computer, using USB 2.0 and USB 3.0 ports, respectively. (B) Photograph of the open illumination dome, comprised of two half-domes; a specimen is mounted inside. Two LEDs are mounted on the inside of the half-domes, and are covered by 3D printed diffusers. The slits at the bottom of each semi-sphere allow the pin of the X-axes stepper to move, as indicated in (C). (C) Side-view drawing of the assembled scanner, illustrating the range of motion of the X, Y and Z axes. The Y-axis stepper is unlimited. Additional photographs of the assembled scanner, as well as links to the parts lists and 3D models, can be found in the Supplemental Material.

The scanner consists of three main components (Fig. 1): (i) an illumination dome which ensures flat and diffuse lighting (inspired by the design in Ströbel et al., 2018), (ii) a two-axis gimbal to change the orientation of specimen inside the dome (inspired by the design of Nguyen et al. (2014)), and (iii) an actuated camera slider to alter the position of the focal plane.

(i) The illumination dome is comprised of two semi-domes (Fig 1B). As the image background needs to be as uniform as possible to achieve high-quality results in subsequent masking and reconstruction steps (see below), the inside of both semi-domes was sanded down, and coated in a matt light grey spray paint, so ensuring flat and diffuse lighting, regardless of specimen orientation. The dome is illuminated by two arrays of LEDs. Accurate colour information requires LED strips with a Colour Rendering Index (CRI) of ≥90 and a colour temperature of ≥5,000 K. In order to reduce sharp specular reflections, translucent, 3D-printed rings are mounted in each half of the illumination dome and act as diffusers (Fig. 1B). Further specifications, as well as links to the parts list and 3D models, can be found in the Supplemental Material.

(ii) Inside the dome, the specimens are mounted directly on a rod connected to the two-axis gimbal. The horizontal (X) and vertical (Y) axes of the gimbal are actuated by two 4-lead NEMA 17 stepper motors (0.7 A, 1.8° step size). In order to minimise the noise introduced by motor jitter, a counterweight is attached to the top of the gimbal, opposite to the vertically oriented stepper motor actuating the Y-axis. The gimbal is mounted such that the X and Y axes run directly through the centre of the illumination dome, enabling specimen rotation without translation relative to the imaging plane. Mirroring slits on the underside of the dome provide the space required for the upwards facing pin of the gimbal to move freely (Figs. 1B and 1C).

(iii) To extend the depth of field per XY position, the camera is mounted on a linear slider (Z axis, Fig. 1A), so that the focal plane can be moved relative to the specimen. The stepper-controlled camera slider is mounted on top of an elevated platform, manufactured from 4 mm acrylic sheets via laser cutting, and is placed outside the illumination dome such that the lens points at the centre of the dome (Fig 1C). We used a 20 MP colour sensor camera (BFS-U3-200S6C-C: 20 MP; FLIR Systems, Wilsonville, OR, USA) with custom made extension tubes and a compact 35 mm, F16 MPZ Computar lens (Computar, CBC Group, Phoenix, AZ, USA). scAnt also supports DSLR cameras, and support for other cameras may readily be added (see GitHub as well as Thingiverse).

All communication between the computer and electronic components of the scanner occurs directly via USB to alleviate the need for additional control hardware. All three stepper motors are controlled by separate USB driver boards (Tic T500; Pololu Corporation, Las Vegas, NV, USA). In order to minimise the number of cables and ports running to or being used at the computer, respectively, all stepper drivers are connected to a USB Hub; a single 2.0 connection from the hub to the computer is then sufficient for all communication apart from camera control. The camera requires higher bandwidth and is therefore connected via a USB 3.0/3.1 port (Fig. 1A).

All stepper drivers, as well as the LEDs, are connected to a generic DC12V, 5A power supply. Due to the low load required to actuate the gimbal, the four-wire steppers do not draw a current close to the peak of either driver or power supply, so that the voltage remains approximately constant. In order to further reduce the peak current draw, steppers are always actuated successively rather than simultaneously.

The stepper drivers controlling the X- and Z-axes are connected to a limit switch, which provides a reference position, and prevents gimbal and camera from moving to positions where they may physically interact (Figs. 1A and 1C). The system is constructed such that the object of interest can be scanned from any orientation about the Y-axis, and an angular range of 100° about the X-axis (Fig. 1C). The range of motion in the X-axis is limited to ensure that the back half of the illumination dome is the background throughout the imaging range. Viewing angles larger than 100° in the x-axis angles are not necessarily required to generate high-quality models as long as the dorsoventral axis of the specimen is approximately aligned with the Y-axis: multiple images obtained for increasingly dorsal or ventral views then contain redundant information, due to the lateral symmetry of insects in the sagittal plane.

The physical dimensions of the scanner impose an upper size limit of approximately 8 cm for the longest specimen axis, whereas the lower size limit depends on the desired resolution, and can be controlled via an appropriate choice of camera and lens. Imaging larger specimens requires a geometric scaling of the scanner, which simple, as all software components and subsequent processing steps remain identical. Hence, scAnt can be readily adjusted to suit the specific needs of the end-user.

Mounting pinned specimens

Photogrammetric reconstruction requires that the appearance of the captured subject is independent of the perspective (Mathys, Brecko & Semal, 2013; Galantucci, Guerra & Lavecchia, 2018). As a consequence, lighting must be as uniform as possible, and any motion of parts of the imaged object relative to each other needs to be avoided. Practically, this means that insect specimens should be dried prior to imaging.

In preparing a specimen for scanning, we recommend posing it with its legs and antennae spread in the frontal plane to minimise occlusion (see Walker et al. (1999) for an in-depth review of the handling and pinning of arthropods). Specimens need to be connected to a pin, which can be glued onto the rod connector of the gimbal; we recommend UV glue as it is easy to remove, but other solutions, such as attaching additional clamps, are possible.

Automated image capture and processing using the scAnt GUI

In order to automate the scanning process as much as possible, we developed a GUI in Python, which provides full control over all relevant settings. Key Python libraries which power the GUI include NumPy (Oliphant, 2006), scikit-image (Van der Walt et al., 2014), OpenCV (Bradski, 2000) and Pillow (Clark, 2015). The GUI consists of five sections (see Fig. 2): (i) The video stream of the camera is shown in the “Live-View”-section of the GUI. Overexposed areas, as well as a normalised histogram for each RGB colour channel, can be displayed as an overlay to aid the selection of suitable settings. (ii) All relevant camera parameters such as exposure time, gain level and white balance ratios can be adjusted in the “Camera Settings”-section. In addition, initial exposure- and gain levels can be determined automatically with a custom-written function. All settings, such as gain level and exposure time, stepper positions and processing options, can be saved as presets to be reloaded for subsequent scans. (iii) The “Scanner Setup”-section is used to configure the project output location on the connected PC. Defined presets of all relevant scanning parameters can be saved to be re-used at a later stage. The scan is also started or stopped from this sub-window. Simultaneous stacking and masking of captured images can be enabled, which may reduce the total time required for a single scan, provided that sufficient computational power is available. Due to the size of the uncompressed images, processing speed does not only depend on CPU core count and clock speed, but also on reading and writing speeds of the hard drive on which the output is stored. (iv) The “Stepper Control”-section allows users to move the scanner to any position within the available range, to home or reset the three axes, and to define the number of images to be taken, determined by the minimum and maximum position for the three axes, and the corresponding step size. These parameters generally vary with specimen size and may be adjusted accordingly. (v) The “Info”-section displays the progress of the scan, information about the status of the scanner, a timestamp.

Figure 2 Graphical User Interface of scAnt.

The Live View (maroon) displays the camera video stream. Bright-red areas indicate over-exposed pixels to aid the adjustment of the camera settings; a histogram of the RGB colour channels is displayed in the lower right corner. All relevant Camera Settings (red) such as exposure time and gain can be adjusted, the live view can be started, and images can be captured. The Scanner Setup-section (yellow) allows users to define an output folder, load existing project configurations (“presets”), write the scanner configuration to the specified output folder, choose stacking and masking methods, and start/abort the scanning process. The Stepper Control-section (navy) is used to change and configure the positions of the camera and mounted specimen. The position limits for the scanning process are also defined in this section. The X- and Z-axes (see Figs. 1A and 1C) of the scanner can be returned to their home positions, and the zero-position of the Y-axis can be reset. The Info-section (blue) displays the progress of the current scan, and all events logged by the scanner in chronological order.

Image processing

The images recorded during a scan are processed in three successive steps: (i) A single EDOF image per perspective is produced by stacking all images per unique XY position; (ii) each EDOF image is masked to remove the image background; (iii) meta-data required for 3D reconstruction, such as sensor size, focal length and camera model, are written to the EDOF image files. All image processing steps are either implemented directly in Python or called from within Python, if they are based on pre-compiled software packages (details below). All processing scripts can be run from within the scAnt GUI, but are also available as standalone Python files available on our GitHub page (https://github.com/evo-biomech/scAnt).

Generating EDOF images

As the depth of field is typically small compared to the dimensions of the specimen normal to the image plane, multiple images per unique XY position are required to capture every part of the body in focus (Fig. 3A). These multiple images are then processed into a single EDOF image, using a series of processing steps (Fig. 3B). First, images are aligned to correct for minor movements in the XY-plane of the camera rail. Second, the aligned images are blended, resulting in a single EDOF image on which all body parts are in focus. Image alignment is performed with Hugin (by Pablo d’Angelo, License: GNU GPLv2), an open-source toolbox for panorama photo stitching and High dynamic range (HDR) merging. High-quality merging requires to exclude all images which do not contain any relevant in-focus pixels. As it is not feasible to programme the scanner such that imaging is automatically constrained to relevant imaging planes, some images will fall into this category. For convenience, these images are automatically removed prior to the stacking process. The variance of a Laplacian 3 × 3 convolutional kernel is computed for each image, and images with a variance below a specified threshold are discarded. The optimal threshold depends on image noise, Z-axis step size, aperture and magnification, and hence has to be determined empirically; we provide a set of suitable parameter combinations as configuration presets (see GitHub). The selected images of a stack are then passed to Hugin’s align_image_stack function with a set of modifiers (see Table 1).

Figure 3 Generation of Extended Depth Of Field (EDOF) images and masks.

(A) Image capturing: Images with different focal planes are automatically captured by moving the camera along the Z-axis for each unique XY position (perspective). Between 20 and 50 images for each perspective are usually sufficient to cover the extent of a specimen of typical size normal to the imaging plane. The Leptoglossus zonatus specimen shown in (A) is captured at a step size of 18° about the Y-axis, but higher spatial resolutions can be configured in the GUI. (B) Image stacking: the images of each stack are aligned, and focus masks are generated to combine the sharpest regions of each image into a single EDOF image. (C) Image masking: These EDOF images are then “masked” to remove the image background. Noise is removed by applying a 5 × 5 Gaussian kernel, and edges are enhanced using Contrast Limited Adaptive Histogram Equalisation (CLAHE) (Zuiderveld, 1994). Subsequently, the largest contour within the image is extracted using a random forest-based edge detector (Dollar & Zitnick, 2013). The resulting outline may include unwanted areas (highlighted in red) which are removed using adaptive thresholding. The noise introduced by the thresholding process is reduced with Connected Component Labelling (CCL) to produce a clean image mask (Fiorio & Gustedt, 1996; Wu, Otoo & Shoshani, 2005). (D) Image cut-out: The generated mask is applied to the EDOF image to create a single cut-out image per XY-orientation.

Table 1 List of modifiers required for image stack alignment.

Modifier	Effect	
-m	Use the field of view of the most distal captured focal plane as a reference, and align and rescale all successive images accordingly (required as the magnification of successive images increases as the camera moves towards the object; more sophisticated methods are available, provided that the exact position of the camera relative to the scanned object is known, see Ströbel et al., 2018).	
-x -y	Align each successive camera view in both x- and y-axes.	
-c 100	Set the number of extracted control points to 100. The number of control points affects the accuracy of the correction to image pitch and yaw, as well as of the lens distortion. While more control points generally increase the accuracy of the alignment process, the processing time increases exponentially. In some cases, excessive numbers of extracted control points can even decrease alignment quality due to inaccurate matches, amplified by shallow focus overlap between images.	
--gpu	Force the use of the GPU for remapping (optional)	

Aligned images are exported to a temporary folder and subsequently merged into a single EDOF image using Enblend-Enfuse (licenced under GNU GPL 2+), initially developed by Andrew Mihal and now maintained by the Hugin developer team. As for image alignment, we provide a pre-defined set of modifiers to Hugin’s enfuse function (see Table 2), chosen to represent a sensible compromise between quality and processing time. Most modifiers alter luminance control, owing to the history of Enblend-Enfuse, which was developed to perform exposure correction for HDR image processing.

Table 2 List of modifiers required for image stack merging.

Modifier	Effect	
--exposure-weight=0	Determines the contribution of pixels close to the ideal luminance of the blended image. A value of zero implies that all pixels contribute equally.	
--saturation-weight=0	Determines the contribution of highly-saturated pixels. A value of zero implies that all pixels contribute equally.	
--contrast-weight=1	Determines the contribution of pixels with high local contrast, which result from sharp edges. A value of one amplifies the contribution of these pixels.	
--hard-mask	The use of hard-masks increases the level of detail in the final image and reduces halos where the outlines of focal planes overlap, as it uses only information from the sharpest focal plane	
--contrast-edge-scale=1	Determines the pre-processing function used prior to edge detection. A value of unity uses local-contrast-enhancement.	
--gray-projector=l-star	Determines the relative weight of the colour channels for greyscale conversion. By default, the colour channels are averaged, but if halos are visible in the EDOF image, 1-star can be set as the grey- projector instead (resulting in an emphasis of small contrast variations in highlights and shadows). The 1-star conversion is disabled by default, as it is more computationally expensive, but it can be activated within the scanner setup section of the GUI under the option “stacking method” (Fig. 2).	

Image masking

The background of the EDOF images is removed in a masking step, which noticeably increases the quality of the mesh created during 3D reconstruction for at least three reasons. First, incorrect matching of background-features in the initial camera alignment step of the reconstruction process is avoided. Second, the number of “floating artefacts” is reduced. Third, the contours of the resulting model are retained more accurately. Masking is hence supported if not required by most of the existing photogrammetry software. The masking process is conducted in a sequence of five steps (see Fig. 3C): (i) Enhancing contours of the EDOF image, (ii) approximating the specimen’s outline, (iii) removing superfluous infill, (iv) cleaning of the generated binary mask, and (v) applying it to the input image. This process is of comparable accuracy to alternative high-quality masking methods such as the use of backlighting (Ströbel et al., 2018; Li & Nguyen, 2019), but has the advantage of significantly reducing capturing and processing times, as well as the required hard-drive space (see Supplemental Material).

(i) The contours of each EDOF image are enhanced by increasing the local image contrast via Contrast Limited Adaptive Histogram Equalisation (CLAHE) (Zuiderveld, 1994). Subsequently, (ii) contours within each EDOF image are identified using a pre-trained random forest-based edge detector (Dollar & Zitnick, 2013), which extracts the outline of the largest shape in the image—the specimen. In order to reduce the detector’s susceptibility to noise, Gaussian and median blurs are applied to each image prior to edge detection. Compared to a Sobel filter alone, the random forest-based edge detector returns more coherent outlines at comparable inference times, which is favourable when extracting the single largest shape in a given image (Bradski, 2000). The resulting outlines separate the specimen from its background, but they may include unwanted background areas, such as the area enclosed between a leg and the main body (Fig. 3C). (iii) These incorrect assignments are removed with adaptive thresholding (akin to chroma- or luminance-keying to remove a specified colour region from an image), which works well as long as the lighting of the image centre is relatively uniform. (iv) In order to identify both incorrectly retained or removed areas of a size below a threshold determined by the image resolution, the mask is then cleaned, using connected component-labelling (Fiorio & Gustedt, 1996; Wu, Otoo & Shoshani, 2005). Finally, the resulting mask of the EDOF image is exported as a binary *.png file, in which white areas represent parts of the specimen, and black areas represent the background. (v) Not all photogrammetry software natively supports the use of masks, and additional cut-outs can be generated by applying the binary mask to the input EDOF image, either as an alpha channel or simply by setting all background pixels to a value of zero.

Adding metadata to processed images

In a final step, all relevant metadata is written into the generated EDOF image, using the open-source tool ExifTool (v. 12.00, developed by Phil Harvey, licenced under GPLv1+). Accurate 3D reconstruction relies on setup-specific image metadata, such as the camera’s sensor width and the focal length of the lens, to “undistort” the EDOF images, match features between images, and approximate the dimensions of the scanned object. The parameters are saved automatically (see Table 3), and can be readily adjusted (the required parameters depend on the reconstruction software, see config.yaml in the Supplemental Material on GitHub).

Table 3 List of key metadata parameters required for the photogrammetry process.

Parameter	Value example	Effect	
Make	FLIR	Required for correct assignment in the camera sensor database	
Model	BFS-U3-200S6C-C	
CameraSerialNumber	XXXXXXXX	All EDOF images must share the same CameraSerialNumber, so that the same camera intrinsics, solved once for a single scene, can be applied to every scene	
FocalLength	35.0	Required to compute the correct magnification and distortion of each image. The FocalLength parameter refers to the value provided on the lens.	
FocalLengthIn35mmFormat	95.0	This parameter refers to the equivalent focal length on 35 mm film, and therefore depends on the sensor width of the camera.	
SensorWidth	13.1	This parameter is required by some meshing software to correctly compute the camera intrinsics in addition to the FocalLengthIn35mm-parameter.	

Reconstruction

The result of a complete scan is a set of masked EDOF images from multiple perspectives. These images now need to be combined to reconstruct a 3D model of the imaged specimen. Numerous photogrammetry software packages are available for this task, be they commercial (e.g. 3DSOM (CDSL Ltd., London, UK), Metashape (Agisoft LLC, St. Petersburg, Russia), 3DF Zephyr (3DFLOW, Verona, Italy)), or open-source (e.g. Meshroom (AliceVision, 2018)). All these software packages have in common the stages in which a mesh is generated from the aligned images: (i) extraction of image features; (ii) matching extracted features between images; (iii) generating structure-from-motion (SfM), structure-from-silhouette (SfS), or a combination thereof; (iv) meshing and mesh-filtering; and (v) texturing. As a consequence, the procedure described below is to a large extent independent of the software choice. SfM reconstructs 3D models by matching extracted features, or “descriptors”, between images; additional information such as data from accelerometers, gyroscopes, or GPS may aid in solving the “camera motion”. The resulting triangulated features, or “sparse point clouds”, form the basis for reconstructing the three-dimensional geometry of the captured scene, and are often combined with approximated depth maps (Seitz et al., 2006; Jancosek & Pajdla, 2010). In contrast, SfS instead uses silhouette, or “masked”, images to produce a visual hull (Laurentini, 1994; Jancosek & Pajdla, 2011; Nguyen et al., 2014). SfM is particularly powerful when entire scenes are to be reconstructed, whereas SfS plays out its strength in the reconstruction of single objects in somewhat “sterile” conditions. We choose SfM, as it outperforms SfS in retaining structural detail such as concave shapes within the model that cannot be captured by image masks alone (Atsushi et al., 2011; Nguyen et al., 2014).

We did the bulk of our work with the open-source solution Meshroom 2020, to maximise the accessibility of scAnt, and hence focus the following description on the reconstruction process with Meshroom. In order to demonstrate both the competitiveness and limitations of open-source software, we reconstructed some models with the “lite” version of the commercial software 3DF Zephyr V5.009 (~€120).

In the first reconstruction step, camera perspectives are aligned based on their visual content, which is represented by extracted feature vectors. In (i) feature extraction with descriptors such as SIFT (Lindeberg, 2012), ORB (Wang et al., 2015) and AKAZE (Alcantarilla, Bartoli & Davison, 2012; Tareen & Saleem, 2018), there is a stereotypical trade-off between the number of descriptors (which determines the accuracy of the camera position matching and sampling density of triangulated features in subsequent steps) and the resulting processing times. Higher sampling densities consistently lead to higher model quality, and in particular, excel at retaining smaller structures such as antennae or thin leg segments. However, as long as each camera perspective is matched correctly in the subsequent step, the number of extracted features does not need to be excessively large. As the SfM calculations are based on the assumption of a constant scene and a moving camera, the background needs to be excluded using the generated masks. In Meshroom, image masks need to be applied directly to the input EDOF images prior to loading the “cut-outs” into the software (Fig. 3D). A detailed description of the key parameters which require adjustment within Meshroom can be found on our GitHub page (see also Supplemental Material).

(ii) After extracting feature vectors from each camera perspective, they are matched, i.e. the extracted feature vectors are compared to propose a set of likely neighbouring views. (iii) These view-pairs are then directly compared to reconstruct the location and rotation of each camera perspective iteratively, and to triangulate extracted features—the core process of SfM reconstruction. The quality of the camera alignment is crucial for the final mesh quality, but is strongly dependent on the number and quality of the previously extracted features. As a consequence, the reconstruction may fail, particularly when scanning small objects or objects which are visually similar across multiple perspectives. In order to address this problem, we have written a Python script which computes an approximated SfM reconstruction of the camera positions to serve as a starting point. This script, estimate_camera_positions.py, takes the project configuration file generated during the scanning process as input and produces a *.sfm file with the solved camera intrinsics and transformation matrices for all EDOF images. This file can then be loaded into the open-source software Meshroom v2020 or later (Fig. 4).

Figure 4 Resulting camera positions of a completed scan.

(A) Approximated camera positions computed from the settings provided by the project configuration file and the scanner dimensions. A transformation matrix for each camera is calculated and displayed as a vector (blue). The initial camera position is highlighted in red. The black dot in the centre represents the scanned object (B) Solved camera positions of the structure-from-motion (SfM) reconstruction performed with Meshroom.

Camera perspective alignment may also be improved through the use of a textured reference object (Ströbel et al., 2018). Although this approach can increase the quality of the alignment, the calibration is uniquely tied to the specific imaging positions used during the scan. In other words, the system would need to be recalibrated every time the step sizes are changed. In contrast, computing approximated positions from a new project file only takes a few seconds.

The result of the SfM reconstruction is a set of aligned camera positions, as well as a “sparse point cloud” of the scanned specimen. (iv) On the basis of this information, a depth map is then computed for each camera view, and a cohesive mesh is calculated. We set the meshing parameters to the highest levels of detail, favouring retention of small features over noise-reduction and smoothing. An in-depth explanation of the meshing process within Meshroom can be found in the literature (Boykov & Kolmogorov, 2004), Labatut & Keriven (2009) and Jancosek & Pajdla (2010, 2011, 2014). Several mesh-filtering steps, specific to Meshroom, are then applied to the generated mesh:Mesh Filtering is primarily used to remove unwanted elements of the resulting mesh by defining a size-threshold. Due to the masking process (see above), the background noise is minimal, so that the number of excluded elements is usually small.

Mesh Denoising improves mesh topology and is best applied prior to Mesh Decimation or Mesh Resampling. Rather than reducing detail across the entire mesh, Denoising smoothens the input mesh across large surfaces, while leaving sharp edges intact, so decreasing noise without simplifying the mesh’s overall topology. Meshroom’s implementation of de-noising is based on the filtering methods described in Zhang et al. (2019).

Mesh Resampling can be used to reduce the number of vertices while retaining the overall shape, volume, and boundaries of the mesh. In contrast to Mesh Decimation, i.e. the deletion of redundant vertices, the mesh is rebuilt entirely using elements of predetermined dimensions. Generally, only one of the two methods should be used.

Last, (v) an image texture is generated by projecting the colour channel information of all undistorted camera views back onto the model. Depending on the size of the model and the number of faces, different unwrapping methods may be used to fit the colour information into a single image texture. We used the Least Squares Conformal Maps unwrapping method (Lévy et al., 2002) to produce image textures of 4,096 px * 4,096 px for meshes with less than 600,000 faces. For larger meshes, we used mosaic texturing. In our trials, LSCM performed consistently better than mosaic texturing methods in retaining colour relationships between neighbouring regions. In addition, the resulting model textures can be edited more easily in subsequent manual post-processing steps.

We accurately replicated the lighting and camera setup of the scanner in Blender v2.8 to compare the quality and accuracy of the model to the original images. We also matched the position of exemplary EDOF and masked images, in order to demonstrate the absence of distortion and the level of detail retained both in topology and texture (see Supplemental Material).

3D model post-processing and rigging

All manual post-processing was performed with Blender v2.8. The mesh of the mounting pin was removed, either by selecting the vertices connecting to the insect mesh and removing all connected faces, or by a simple Boolean intersection. Subsequently, the resulting hole was filled by collapsing the surrounding vertices to their centre. If desired, basic or Laplacian smoothing modifiers, as well as manual smoothing via sculpting can be used to improve the local mesh quality (Fig. 5A).

Figure 5 Mesh post-processing steps.

(A) Mesh cleaning process (here for Pachnoda marginata). (i) The mounting pin and any floating artefacts are removed from the generated mesh, and (ii) the topology is cleaned, using Laplacian smoothing, before (iii) producing the final mesh. (B) Thin, planar structures, such as some wings, are difficult to reconstruct but may be added to the models by approximating them as 2D objects. (i) Original, generated mesh (Sympetrum striolatum). (ii) cut-outs of each wing are taken from masked EDOF images. (iii) The cut-outs are used as image textures for planes of the corresponding size, containing an alpha layer, and are merged with the cleaned mesh. (C) Rigging process. Insets show close-ups of the legs and abdomen at each stage of the process (Sungaya inexpectata). (i) Reconstructed mesh pose as-scanned. (ii) Assigned armature superimposed over the untextured mesh. Joints are placed at each abdomen intersection as well as at every identified joint location to create a fully articulated model (in principle, the number of joints will be well defined for body parts such as legs, but may be harder to define for body parts such as antennae, where an appropriate number of joints may be determined by the desired flexibility and degree of accuracy for the posing). (iii) After assigning the surrounding vertices to their respective rigid bodies, the model can be posed arbitrarily. As an illustrative example, we extended the curled abdomen and posed the legs to reflect the posture of a freely standing stick insect.

Thin and transparent structures, such as wings, are challenging for SfM-based reconstruction and often appear fragmented (Fig. 5B and see Nguyen et al., 2014; Ströbel et al., 2018). However, structures such as wings may be approximately described as two-dimensional, provided that their thickness and curvature is small compared to the resolution of the scans. In such instances, there exists an easy yet high-quality option for reconstruction: wings can be added to the model during post-processing, by re-projecting the wing structure from EDOF images onto image planes, using the corresponding mask as an alpha-channel (Fig. 5B). An orthogonal top-down view of the fragmented structure or sparse point cloud of the scanned specimen serves as the wing outline. The cut-out structure is then manually transformed in scale and rotation to match the original structure, using an image editing program such as GIMP (licenced under GPLv3+); the scaled wing is then set as the image texture of a plane object with corresponding dimensions. By blending a diffuse shader for the colour information with a transparent shader for the opacity, the wings are added to the textured mesh, using the image plane’s alpha channel as an input. Subsequently, all unconnected vertices of the fragmented wing are removed, using the process outlined above for the removal of the mounting pin. This process retains key visual information of the wing such as colour and venation patterns, but of course does not correspond to a “true” 3D reconstruction, as the wing is simplified as a planar structure. This approximation may be valid in some cases (see Fig. 6, Orthomeria versicolor), but will correspond to a significant simplification in others. For a better overview of the resulting model quality, refer to our Sketchfab Model collection.

Figure 6 Examples of fully textured 3D models created with scAnt.

All models have been reconstructed with either Meshroom 2020 or 3DF Zephyr Lite (Table 4), and were post-processed and rendered in blender v2.8, using cycles. Boxes on the right illustrate various quality criteria for model reconstruction (A) Ventral, sagittal, anterior, and posterior view of Porcellio scaber, demonstrating the achievable model quality for the smallest scanned specimen. (B) Front right tarsus and claws of a Dinomyrmex gigas specimen, illustrating the mesh density, as well as the texture quality at small scales. (C) Rigged, untextured mesh after the pin has been removed and mesh smoothing has been applied to the model. (D) and (E) Close-ups of the colour detail retained for the elytra and thorax of the reflective Metallyticus splendidus specimen. Topology and texture detail of a Sungaya inexpectata model reconstructed with Meshroom (F) and 3DF Zephyr Lite (G). Zephyr captures fine surface texture better than Meshroom. All models, as well as additional details on SfM reconstruction and post-processing steps can be found on Sketchfab.

Table 4 Overview of scanned specimens.

Species	Length of longest axis (excluding antennae and legs) (mm)	Authority	Reconstructed using Meshroom (M), 3DF Zephyr Lite (Z)	
Amphipyra pyramidea	20.4	Linnaeus 1758	M & Z*	
Atta vollenweideri	9.5	Forel 1893	Z*	
Blatta orientalis	27.5	Linnaeus 1758	Z*	
Diacamma indicum	9.3	Santschi 1920	Z*	
Dinomyrmex gigas	22.6	Smith 1858	Z*	
Leptoglossus zonatus	19.4	Dallas 1852	Z*	
Metallyticus splendidus	30.0	Westwood 1835	M, Z*	
Myathropa florea	14.8	Linnaeus 1758	M*, Z	
Orthomeria versicolor	37.5	Redtenbacher 1906	M*	
Pachnoda marginata	26.7	Kolbe 1906	M*	
Porcellio scaber	7	Latreille 1804	M, Z*	
Sungaya inexpectata	36.3	Zompro 1996	M*, Z*	
Sympetrum striolatum	28.3	Charpentier 1840	Z*	
Notes:

* Indicates the mesh shown in Fig. 6 (side by side comparisons available on Sketchfab).

The length of the longest body axis is measured on the final mesh within Blender v2.8. When both Meshroom (M) and 3DF Zephyr (Z) are listed in the right column, separate meshes were created from the same masked EDOF images for comparison.

In a final step, the mesh may be rigged, by which we mean the assignment of virtual joints and rigid body parts, subsequently allowing users to pose the models (Fig. 5C); this process is possible in arthropods, because their bodies are reasonably approximated as a linked series of rigid bodies. As long as joint-location and type can be defined with reasonable accuracy, rigging allows for anatomically correct posing. In order to rig a scan, we create an armature inside the mesh and define joint types and locations. In order to ensure that labelled segments are treated as rigid bodies, we use weight painting, which avoids incorrect deformation arising from smoothing algorithms which deform tissue locally.

In order to demonstrate that rigging of 3D models is not only aesthetically pleasing but also scientifically useful, we asked seven individuals of varying degree of experience (undergraduate students, PhD students, Postdocs) to measure a set of three anatomical parameters of a specimen of Sungaya inexpectata: Head width (HW), femur length (FL, right front leg), and abdomen length (AL, see also Ströbel et al., 2018). These measurements were taken either on the physical specimen, using a LEICA Z6 Microscope (Leica Camera AG, Wetzlar Germany), in the following referred to as a “2D measurement”, or on the final 3D model using the internal measuring tool in Blender v2.8. In both cases, participants were allowed to rotate the specimen and choose their preferred magnification freely. We included HW and FL as they are relatively easy to measure and unlikely to be subject to large parallax error. As a consequence, we expect the difference between measurements from 2D vs 3D data to be small. In contrast, the abdomen of the original specimen was curved in two axes (Fig. 5C); we removed this curvature via rigging, straightening the abdomen to a position similar to that observed in live animals, so enabling us to test if the flexibility provided by rigging can help to increase measurement precision.

Accuracy evaluation

In order to quantify the reconstruction accuracy achievable with scAnt, we performed a set of measurements on certified gauge blocks, as described by Gallo, Muzzupappa & Bruno (2014). Grade 0 gauge blocks (uncertainty of ± 0.00008 mm, UKAS certified) of 1.50, 1.10, 1.05, and 1.00 mm were scanned in pairs in a 3D printed tray with a rectified surface to create step-sizes of 500, 100 and 50 µm, respectively. The gauge cubes were then reconstructed using the parameters described in the Meshroom Guide on our GitHub Page. The step-sizes were measured in Blender v2.8, using a custom-written Python script. In brief, we measured the average vertical distance between all vertices of the top plane of the step cube (1.50, 1.00 and 1.01 mm) and the 1.00 mm reference gauge cube.

Scanned species

In order to demonstrate the versatility and quality of scAnt, we created 3D models for a series of arthropods, selected to present various challenges such as size, transparency, hairy surfaces, or iridescence (see Table 4; Fig. 6, Sketchfab).

Results and discussion

We created several models to demonstrate the quality achievable with scAnt (see Fig. 6 and Sketchfab). The reconstruction error, as estimated from the reconstruction of pairs of grade 0 gauge cubes, is approximately 12–15 µm across step sizes (see Table 5). In the following, we briefly address some key features of these models to demonstrate their quality, highlight key challenges and difficulties which users may encounter, and outline the current performance limits of scAnt. We then proceed to describe how 3D models generated with scAnt may be used in research, outreach and conservation.

Table 5 Distance measurements on certified gauge cubes.

Step-size (mm)	0.500	0.100	0.050	
Measured mean (mm)	0.517	0.112	0.065	
Std (mm)	0.005	0.006	0.005	
Relative error	3.45%	12.67%	30.83%	
Note:

The reported measured mean refers to the distance between the vertices of the top plane of the step-cube (1.50, 1.10 and 1.05 mm respectively) and the reference cube (1.00 mm).

Quality, challenges and limitations

The quality of the final models is determined by the quality of the EDOF images and the reconstruction process. The former hinges on the quality of the used camera and lens (to first-order; image processing during stacking introduces second-order effects on quality); high-quality cameras and lenses are available where the budget allows, and the resolution and scanner dimensions can easily be adjusted to specific needs. Hence, model quality is mostly limited by the quality of the 3D reconstruction, for which we identified four major challenges.

First, there exists a lower size limit for features that can be retained during reconstruction. The smallest features that can be retained depend on the resolution of the input images, the quality of the generated masks, the number of camera positions, the quality of meshing algorithms and—crucially—the size of the scanned animal (Fig. 6A): The larger the animal, the coarser the absolute resolution. As an illustrative example, the thinnest preserved structures of the Dinomyrmex gigas specimen with a body length of 27.5 mm are the claws, with a width of approximately 30 µm, or about three times the accuracy of the reconstruction (they are likely even thinner towards their tips; Figs. 6B and 6C). The choice of reconstruction software also notably contributes to the smallest retained features; Using the same input images, the low-cost commercial software 3DF Zephyr Lite generated a higher level of topology detail than the open-source option Meshroom (see Figs. 6F and 6G), and thin segments such as tarsi were less likely to be fragmented. We provide a number of direct comparisons between the two software options on Sketchfab.

The reconstruction of small features is particularly challenging where large size differences between feature and body size are combined with high local curvature, such as for the proboscis of Leptoglossus zonatus (Fig. 6, second row and see Sketchfab as well as Supplemental Material). The reconstructed proboscis was partially fragmented, which required manual post-processing in the form of manually joining the segments into a coherent mesh with Blender. Generally, structures with high-aspect ratios, such as hairs, may be visible in EDOF images, but fail to reconstruct. The most straightforward way to address this issue is to increase the number of camera positions during the scan, which increases the number of matched descriptors, but comes at the cost of longer processing times. In order to counter this increase in processing time, orientations near the extreme X-axis angles could be sampled more sparsely: an approximately constant angular distance between neighbouring views would maximise the unique information per image, and reduce the total number of images for the same spherical coverage Nguyen et al. (2014). As SfM-based approaches require a sufficient number of clearly defined features to preserve a structure, increasing the contrast using additional filtering methods such as Wallis filters may also help to address this issue (Gallo, Muzzupappa & Bruno, 2014). Alternative meshing methods, such as Visibility-Consistent Meshing, may retain thin features without the need for manual post-processing (see Ströbel et al., 2018), but are currently only available as an experimental feature in commercial software such as of PhotoScan Pro (Agisoft LLC). The quality of EDOF images may also be improved further by enforcing perspective consistent images, which requires high-accuracy calibration of camera positions during image stacking or re-designing the system as a fixed lens setup (Ströbel et al., 2018; Li & Nguyen, 2019). Such additional stacking and calibration options may be included in future updates of scAnt.

Second, highly reflective surfaces, such as the head capsule of the leaf-cutter ant, Atta vollenweideri (Fig. 6, first row), or of the cockroach, Blatta orientalis (Fig. 6, third row) may ironically lead to a comparably rough mesh topology, as the reflection introduces noise in the reconstruction process. This noise is caused by a combination of small variations in appearance with viewing angle and low surface detail, so reducing the number of features matched between views, and leading to incorrect depth maps. In contrast, the reconstructions of seemingly more challenging reflective and even iridescent structures with high levels of surface detail are far less noise ridden (Figs. 6D and 6E). Noise may be reduced using Mesh de-noising, manual smoothing, or similar post-processing techniques, but the specimen may then appear artificially smooth, and it is difficult to achieve a photorealistic appearance. Two alternatives exist. First, the application of fine powder to the specimen’s surface prior to scanning may reduce surface reflectivity, but it may come at the cost of a less realistic optical impression when the model is viewed in variable lighting conditions. Second, the uniformity of the lighting conditions may be improved further, but some reflections from the camera lens or the dome opening are impossible to avoid.

Third, transparent structures, such as some wings, can be difficult to reconstruct in 3D for at least three reasons. First, because they may not be recognised as part of the body, second, because their appearance varies strongly with viewing angle, and third, because pixels may be wrongly assigned to the dorsal or ventral side of the wing. Two solutions to this problem exist: First, the generated masked EDOF images can be used to re-project the fragmented structures with relatively little effort (Fig. 5B). However, this approach is limited to thin and approximately planar wings, as it is unable to capture any out-of-plane variation in wing morphology. Second, alternative meshing methods such as SfS, or Visibility-Consistent Meshing can be used to enforce the inclusion of such regions, but are currently only implemented in commercial software, such as Agisoft PhotoScan Pro. To provide some perspective, the cost of this software exceeds the costs of the scanner, including camera and lens, by a factor of three.

Fourth, model creation is time-intense. High-throughput model creation requires (i) scripting of the complete reconstruction process, which is possible with most photogrammetry software, including Meshroom; and (ii) a reduction in the total processing time. As an illustrative example, all models in Fig. 6 were created with 180–360 EDOF images recorded at 20 MP resolution; Capturing and simultaneously processing the EDOF images using our GUI takes between 2 and 5 h, depending on the number of camera positions (intel core i7, 8 Core processor at 4.3 GHz, 32 GB DDR4 RAM, NVIDIA RTX 2070 QMax). The reconstruction of 3D models took between 3 and 12 h (intel core i9, 12 Core processor at 4.0 GHz, 64 GB DDR4 RAM, 2 × NVIDIA RTX 2080 Ti). Total model creation time may be reduced by sacrificing resolution (in step size or pixel density), by using more powerful computers, or by further improvements in the algorithms underlying the reconstruction.

Notwithstanding these challenges and limitations, the quality of the generated models is sufficient to enable their use in a number of different activities, including research, outreach, and conservation, as we briefly outline in the following.

Research

Morphometry

The development of increasingly sophisticated comparative and geometric morphometric methods has revived the popularity of morphometry in entomology (see e.g. Tatsuta, Takahashi & Sakamaki, 2018). The immediate appeal of 3D models in morphometric studies is two-fold:

First, 3D models enable measurements which are difficult or even impossible to obtain from 2D images. Such measurements include area measurements of structures with complex shape, or measurements of volumes. For example, it is straightforward to extract body surface area and body volume from our 3D models in Blender v2.8, using its default measurement toolbox and 3D-Print add-on (we assigned a uniform thickness to the wings, equivalent to the thickness of the reconstructed fragmented wing). An ordinary least squares regression of log10-transformed data yields a slope of 0.61 (95% CI [0.33–0.98]), consistent with isometry, and the results from Ströbel et al., 2018 (data extracted with WebPlotDigitzer, slope 0.59, 95% CI [0.49–0.68]). Notwithstanding the ease with which such data can be obtained from 3D models, two sources of error require attention. (i) The relative measurement error, defined as the ratio between the measured quantity and the resolution, is usually small for linear measurements, but it is additive, and can hence become sizeable for area and volume measurements, so reducing statistical power. In practice, this error is nevertheless only relevant for small structures, and is unlikely to bias scaling analyses conducted across multiple orders of magnitude, as there is no a priori reason to assume that it is systematic. (ii) Variations in fractal dimension across specimens may introduce a potential systematic error: area (and volume) measurements may rely on perimeter measurements, which can vary strongly with image resolution (Mandelbrot, 1967, 1982). Arthropods are unlikely to have a large fractal dimension, and occupy only a small range of physical length scales (from atomic dimensions to a few centimetres, say). However, body surfaces are often highly structured, so that a correlation between resolution and estimated area/volume is likely. Methods to estimate fractal dimension exist (e.g. Neal & Russ, 2012), and this issue needs further attention to enable a robust comparison of areas or volumes across scans or species.

Second, 3D models may increase measurement accuracy and precision even for measurements which can in principle be taken from 2D images (see, e.g. Ströbel et al., 2018). Indeed, intra-observer errors can be large even for seemingly straightforward measurements (Viscardi, Sakamoto & Sigwart, 2010). We replicated the finding of Ströbel et al., 2018 that the coefficient of variation may be reduced in some (but not all) cases when measurements are taken from 3D models instead of 2D images (see Table 6). We demonstrated further that the increase in precision is largest for structures which can be re-positioned via rigging in the 3D model (see Table 6 and Fig. 5C).

Table 6 Distances measured on a Sungaya inexpectata specimen.

	Head width (mm)	Femur length (mm)	Abdomen length (mm)	
n = 7	2D	3D	2D	3D	2D (curved)	3D (straight)	
min	3.04	3.17	6.08	6.44	16.58	19.43	
max	3.17	3.22	6.42	6.70	19.78	20.10	
µ	3.12	3.20	6.25	6.57	18.15	19.59	
σ	0.04	0.01	0.10	0.07	1.16	0.20	
cV	0.012	0.004	0.016	0.011	0.064	0.010	
statistic	DAD = 4.85, p = 0.028	DAD = 0.85, p = 0.36	DAD = 12.50, p < 0.0001	
Note:

Measurements have been performed using a LEICA Z6 Microscope on the specimen itself (2D), or blender v2.8 on the generated model (3D) (n = 7 different observers). Listed are the shortest (min) and longest (max) distances, the mean (µ), the standard deviation (σ) and the coefficient of variation (cv). We tested for differences in the coefficient of variation using the asymptotic test for the equality of coefficients of variation from k populations after Feltz and Miller, 1996, implemented in the R-package cvequality (Marwick & Krishnamoorthy, 2019), for which we provide the test-statistic (DAD), and the associated p-value.

As a consequence of both advantages outlined above, morphometric data extracted from 3D models may be more versatile and accurate, increase statistical power, and reduce the ambiguity present in 2D images (Roth, 1993; Cardini, 2014; Gould, 2014; Fruciano, 2016; Buser, Sidlauskas & Summers, 2018; Bas & Smith, 2019 but see Courtenay et al., 2018; McWhinnie & Parsons, 2019). While the use of 3D data is still somewhat limited by the power of available statistical methods for subsequent analyses (Polly & Motz, 2016), increasingly advanced methods are available (Bardua et al., 2019), and 3D morphometry is expected to grow in importance.

Machine learning-based segmentation, detection and tracking

Computer vision and machine learning are quickly finding their way into the standard methodological toolbox deployed in behavioural and kinematic studies of animals (Branson, 2014; Dell et al., 2014; Egnor & Branson, 2016; Robie et al., 2017). Recent advances in the use of pre-trained networks have reduced the quantity of required training data—and hence tedious hand-labelling—substantially (Mathis et al., 2018; Redmon, Farhadi & Ap, 2018; Graving et al., 2019; Pereira et al., 2019; Bochkovskiy, Wang & Liao, 2020). We anticipate that 3D models which are photorealistic at least at low resolutions may further improve the generalisability and precision of these approaches, as they enable the creation of labelled “synthetic datasets” (De Souza et al., 2017; Tobin et al., 2017; Varol et al., 2017; Kar et al., 2019; Mehta et al., 2019; Stout et al., 2019). Combining the power of free software engines such as Unreal or Unity3D with the flexibility of rigged 3D models, we can freely pose and position individuals and groups of individuals, control image background, lighting conditions, image noise and degree of occlusion, so providing the opportunity to create a virtually unlimited variety of labelled training images. As illustrative examples, we provide some animations and still images in the Supplemental Material.

Clearly, the effort required to generate synthetic data may exceed the effort to create sufficient hand-labelled training data if all experiments occur in well-defined and controlled environments. However, we hope that the development of an integrated pipeline to create synthetic training data from 3D models may render network performance robust and accurate enough even in “unseen” (and unpredictable) conditions in the field. Synthetic datasets may be a particularly powerful method to increase the performance of detection networks on complex backgrounds, or in the presence of extreme occlusion/overlap between individuals, as occurs regularly in studies on social insects (Gal, Saragosti & Kronauer, 2020).

Conservation and outreach

High-quality 3D models of arthropods may aid in both conservation and outreach, as they bring with them the following advantages (Nguyen et al., 2014; Galantucci, Pesce & Lavecchia, 2015; Erolin, Jarron & Csetenyi, 2017; Ströbel et al., 2018; Cobb et al., 2019; Brecko & Mathys, 2020):

First, digital models can be made available online, so reducing the need for physical visits to collections, or specimen exchange between collections. Therefore, they maximise collection utility and accessibility, and minimise the risk of specimen damage. Second, in contrast to physical specimens, digital models do not deteriorate. Both points may be particularly relevant for the study and availability of valuable holotypes, provided that the digital models are of sufficient accuracy. Third, 3D models add significant value to online encyclopaedias which currently are almost exclusively populated by 2D images. Fourth, 3D models can be animated, so increasing the information content which can be stored and communicated. For example, we are working on an extension of our pipeline to animate our models directly with kinematic data recorded with live animals during natural locomotion. Such animations may be used in teaching, to demonstrate different gaits, say, but also in computer games or educational TV programmes. Fifth, 3D models may be used in “digital exhibitions”, enabling an unprecedented possibility of visitors to interact with the exhibits, and increasing accessibility of rare specimen.

Conclusion

We introduced scAnt 3D, a low-cost, open-source photogrammetry pipeline to create digital 3D models of arthropods and small objects. The process of 3D model creation is largely automated and can be easily controlled via a user-friendly GUI; all required code, technical drawings and component lists are freely available. We achieve state-of-the-art model quality at a fraction of the cost of comparable systems, paving the way for the widespread creation of near photorealistic 3D models of arthropods to be used in research, conservation, and outreach.

Supplemental Information

Supplemental Information 1 Renderings of the 3D Scanner.

(A) Assembled scanner, excluding wiring. (B) Inside of the illumination dome, with the insect pin (vertical rotation axis) at its centre, and a circular array of LEDs in both halves of the spherical illumination dome. (C) Range of positions for the horizontal axis and camera, as controllable via the stepper-controlled camera slider provided in the graphical user interface of scAnt (see Fig. 2). (D) Photograph of the scanner as built.

Click here for additional data file.

Supplemental Information 2 Laplacian filter applied to an image to highlight in-focus areas.

Prior to image stacking, images which do not include relevant in-focus pixels are discarded in an automated process. (A) Image of a Dinomyrmex gigas specimen; the focal plane is approximately aligned with the hind legs. (B) A Laplacian filter can be used as a simple edge detector, as it highlights areas with a high local variation in intensity. Sharp areas result in greater variance so that the variance of the Laplacian can be used as a suitable scalar proxy for the fraction of the image which is in focus.

Click here for additional data file.

Supplemental Information 3 Comparison between (A) the original EDOF image and (B) the 3D model of a Leptoglossus zonatus specimen.

The lighting of the scanner, as well as the camera parameters were replicated in blender v2.8 and rendered with Cycles to qualitatively inspect the models for evidence of distortion as a result of the reconstruction process.

Click here for additional data file.

Supplemental Information 4 Comparison between masks generated from Extended Depth Of Field (EDOF) images.

(A) EDOF image captured with regular, flat lighting. (B) EDOF image captured with backlighting only. (C) Mask generated from the flat lighting EDOF image via random forest-based outline detection. (D) Hand-annotated ground truth mask generated from the flat lighting EDOF image (A). (E) Mask generated from the backlighting EDOF via backlight thresholding as described by Ströbel et al. (2018).

Click here for additional data file.

Supplemental Information 5 Comparison between masks generated from Extended Depth Of Field (EDOF) images.

31 uncompressed images for the flat lighting and backlighting EDOF images were captured at 20 MP resolution. The masking accuracy was computed by comparing the generated mask of each method to a hand labelled ground truth mask (see Fig. S4). The processing times refer to the combined time of the stacking and masking. The total file sizes refer to the combined amount of hard-drive space required to store all original and processed images.

Click here for additional data file.

The authors are grateful for the kind support and expert input from René Bulla, in particular in respect to 3D modelling and processing of visual data.

Additional Information and Declarations

Competing Interests

Author Contributions

Data Availability

The authors declare that they have no competing interests.

Fabian Plum conceived and designed the experiments, performed the experiments, analyzed the data, prepared figures and/or tables, authored or reviewed drafts of the paper, designed the scanner hardware and implemeted all code, and approved the final draft.

David Labonte conceived and designed the experiments, analyzed the data, authored or reviewed drafts of the paper, and approved the final draft.

The following information was supplied regarding data availability:

All code is available at GitHub: https://github.com/evo-biomech/scAnt

Manufacturing and assembly instructions, as well as component lists, are available at Thingiverse: https://www.thingiverse.com/evobiomech/designs

Example models generated with the described 3D scanner and processing pipeline are available at Sketchfab: https://sketchfab.com/EvoBiomech

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
