# Peer review of "scAnt—an open-source platform for the creation of 3D models of arthropods (and other small objects)"

_PeerJ, doi:10.7717/peerj.11155_

## Round 0.1 · original submission · Minor Revisions

Dear Drs. Plum and Labonte:

Thanks for submitting your manuscript to PeerJ. I have now received three independent reviews of your work, and as you will see, the reviewers raised some concerns about the research. Despite this, these reviewers are quite optimistic about your work and the potential impact it will have on research studying 3D reconstructions of invertebrates. Thus, I encourage you to revise your manuscript, accordingly, taking into account all of the concerns raised by the three reviewers.

Please ensure that your figures and tables contain all of the information that is necessary to support your findings and observations.

Please edit the manuscript for clarity and typos. There appear to be some key references missing.

Please note that Reviewer 2 kindly provided a marked-up version of your manuscript.

Good luck with your revision,

-joe

·

Basic reporting

A well written and concise paper with good literature review and supported materials.
Minor improvements could be done:
- Line 190: remove well in "Note well that..."
- Line 525: "...blender..." should be capitalised.
- Add reference to recent relevant publications such as "Perspective-Consistent Multifocus Multiview 3D Reconstruction of Small Objects" https://doi.org/10.1109/DICTA47822.2019.8946006

Experimental design

The paper identifies a gap in 3D reconstruction of insects and present an open source design/solution that uses low-cost off the shelf components and open source software.

The design is well thought and well tested. It proposes a few improvements including using 3D printing for low-cost and streamline design, removing completely out of focus images, using known camera poses from pan-tilt position, automatic background removal, and 3D model rigging.

There are few things the authors could clarify:
- Sentence starting from line 150 "While an angular range of 100° may appear limiting, steeper viewing angles are not required to generate high quality models...". This is not strictly true. The actual problem is that it is difficult to take images at these extreme angles due to the pan motor getting close to the camera. A problem with the current camera capture setting in the paper is that pan-tilt angle steps are kept constant, causing image positions near the north and south poles of the sphere too close to each other. Changing pan angular step to maintain roughly uniform angular distance between two nearest camera positions (look at a soccer ball) will increase the unique information captured in each image and reduce the total number of images for the same spherical coverage.
- Strobel et al 2018 shows that calibrated image alignment for focus stacking is crucial to the success of 3D reconstruction. However the authors use Hugin software which seems not taking account of calibration for image alignment.
- Automatic background removal has always been challenging as it is quite unreliable due to the complex imaging and geometric properties of specimens. Often this requires additional manual corrections which could be quite time-consuming and defeating the automating purpose. Some validations of the proposed automatic background removal technique (probably versus background lighting as proposed by Strobel et al 2018) will be useful.

Validity of the findings

The design and the results presented in the paper are wonderful. Particularly the authors take great effort to share and document the codes, and the components to be purchased or printed. This is one of the best example how to share research effort and help others to reproduce the work.

Additional comments

Great work!

·

Basic reporting

The manuscript is well-structured and written in professional language. However, I feel that the text can be further improved, since it contains some typos, incorrect punctuation and inconsistent formatting. I have annotated the text in the PDF up to my abilities, but, since this is not my area of expertise, I recommend it to be corrected by an expert. Please note that pages 1-10 of the PDF were annotated more thoroughly, and correct the same issues on the rest of the pages accordingly.
References seem adequate, the differences with the previously created photogrammetry devices are highlighted. Figures are well-done, they clearly illustrate the process of photogrammetry. The only exception is Figure 3, which, I think, should follow the description of the process in the text more strictly, and be provided with a more concise explanation in the figure captions. The 3D models provided in the Supplementary materials are excellent.

Experimental design

scAnt software seems useful and user-friendly. The design with one linear and two rotary axes is compact and reliable. However, there are some comments and questions concerning the construction:

1) It is not very clear what the other end of the z-axis rests on. Considering the plastic base, the long arm, and the fact that the weight of the shafts and the leadscrew hangs on the stepper motor mount frame, without additional support at the end of the z-axis, it will sag and shake when the stepper motor is running. Even a small support at the end of the z-axis will increase the rigidity significantly.
2) The choice of the shaft system as linear guides along the z-axis is questionable. Wouldn’t it be easier to use profile rail guides, to avoid the bother of calibration of the alignment of the shafts?
3) The rigidity of the frame of the pivoting axis x, as well as the base of the structure, also raises concerns, since it is proposed to be made by laser cutting from plexiglass. The rigidity can be significantly increased if it is milled from ISO 2024 aluminum, it will not come out much more expensive. Wear resistance will also increase - acrylic will crack and chip, especially at the edges and in the places of screw fastenings, and more so when it is laser cut, not milled.
4) It would be nice if the authors specified the models of LED illuminators that fit the form factor to their design and have a decent CRI.

Validity of the findings

The proposed photogrammetry pipeline should be of great help to the scientific community and enthusiasts.

·

Basic reporting

This is a wonderfully written, and thoughtfully put together article, on a topic of extreme importance. I can't thank the authors enough for this excellent work, and hope that their work will lead to a renaissance in the use and accessibility of 3D-rendered specimens.

Experimental design

I am not an expert in the computer software components of this project, but the methods are clear and straightforward, the application novel, and the utility unquestionable.

Validity of the findings

The results are clearly presented, and supported by the methods chosen.

Additional comments

This is an excellent article, and I am extremely grateful that the authors have put together this package and hope that it is widely adopted by natural history collections across the globe. I know that I will be sharing this article after its publication with the manager of our insect collection in the hope that we will start digitizing our specimens and making them more available to the public.

---

## Round 0.2 · accepted · Accept

Dear Drs. Plum and Labonte:

Thanks for revising your manuscript based on the concerns raised by the reviewers. I now believe that your manuscript is suitable for publication. Congratulations! I look forward to seeing this work in print, and I anticipate it being an important resource for groups studying 3D reconstructions of invertebrates. Thanks again for choosing PeerJ to publish such important work.

Best,

-joe